# You Only Measure Once:
# On Designing Single-Shot Quantum Machine Learning Models

## Abstract

Quantum machine learning (QML) models conventionally rely on repeated measurements (shots) of observables to obtain reliable predictions. This dependence on large shot budgets leads to high inference cost and time overhead, which is particularly problematic as quantum hardware access is typically priced proportionally to the number of shots. In this work we propose *You Only Measure Once (Yomo)*, a simple yet effective design that achieves accurate inference with dramatically fewer measurements, down to the single-shot regime. Yomo replaces Pauli expectation-value outputs with a probability aggregation mechanism and introduces loss functions that encourage sharp predictions. Our theoretical analysis shows that Yomo avoids the shot-scaling limitations inherent to expectation-based models, and our experiments on MNIST and CIFAR-10 confirm that Yomo consistently outperforms baselines across different shot budgets and under simulations with depolarizing channels. By enabling accurate single-shot inference, Yomo substantially reduces the monetary and computational costs of deploying QML, thereby lowering the barrier to practical adoption of QML.

## 1 Introduction

Quantum computing (Nielsen & Chuang, 2010) has emerged as a promising paradigm for advancing computational capabilities beyond the classical regime. In particular, quantum machine learning (QML) (Cerezo et al., 2022; Huang et al., 2022; Biamonte et al., 2017; Benedetti et al., 2019) seeks to leverage quantum resources for learning tasks such as classification (Pérez-Salinas et al., 2020; Schuld et al., 2021; Liu et al., 2025b; Gong et al., 2024), generation (Khatri et al., 2024), and reinforcement learning (Chen et al., 2020; Liu et al., 2024). Unlike classical machine learning, however, QML inherently involves probabilistic measurement outcomes. To obtain reliable outputs, QML models typically require repeated circuit executions, aggregating many measurement shots to estimate expectation values of observables. This reliance on repeated measurements constitutes one of the fundamental distinctions between classical and quantum machine learning.

As a result, the resource requirements for training and inference in QML are substantial. Since access to quantum hardware is predominantly cloud-based and requires waiting in queues, the monetary cost of usage scales proportionally with the number of shots. In addition, shot repetition contributes significant time overhead: in general only a single quantum processing unit (QPU) is available to execute the circuit, limiting opportunities for parallelization. Given the scarcity of quantum hardware resources, this repetition exacerbates both the financial and computational burden of deploying QML models. From the broader perspective of a machine learning model's lifecycle, the inference stage typically dominates the overall cost (Sardana et al., 2023; Samsi et al., 2023). This observation further motivates the development of shot-efficient QML models, particularly in the inference phase. While several recent works have begun exploring shot-efficient methods in QML and variational quantum algorithms (VQAs) (Phalak & Ghosh, 2023; Kim et al., 2024; Liang et al., 2024), and theoretical concepts of single-shot inference have been proposed (Recio-Armengol et al., 2025b), there remains no clear design pathway for implementing such models. Moreover, the practical implications of a truly single-shot inference QML model, both in terms of hardware usage and experimental throughput, have yet to be systematically studied.

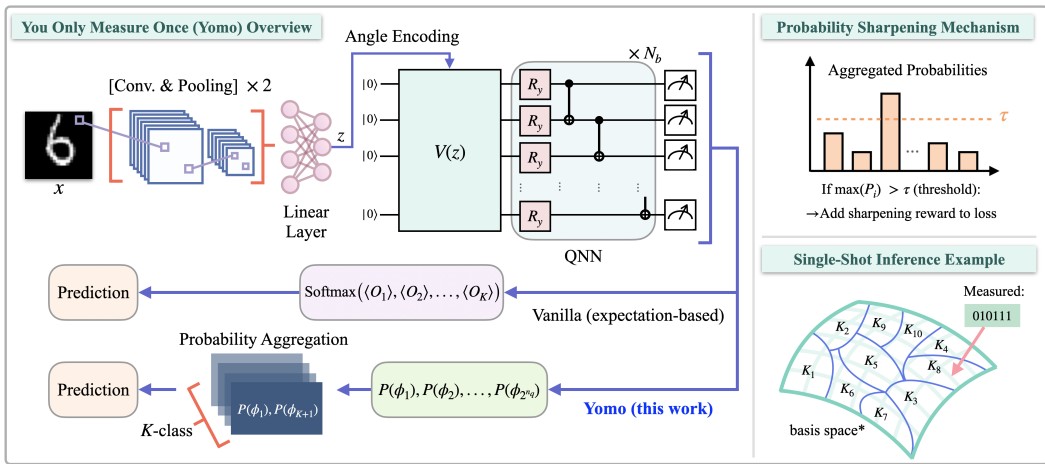

Figure 1: Overview of the proposed Yomo framework. Left: The input image is processed by two convolution–pooling layers and a linear layer before angle encoding into a parameterized quantum circuit (QNN) with $N_b$ layers. The vanilla (Pauli expectation-based) approach applies a softmax over measured expectation values $\langle O_1 \rangle, \ldots, \langle O_K \rangle$ to produce class probabilities. In contrast, Yomo aggregates measurement probabilities $P(\phi_1), \ldots, P(\phi_{2^{n_q}})$ into $K$ classes for prediction. Top-right: Probability Sharpening Mechanism. During training, if the maximum aggregated probability exceeds a threshold $\tau$, an additional sharpening reward is added to the loss, encouraging confident and peaked predictions. Bottom-right: Single-shot inference example. A single measured bitstring is mapped to one of the $K$ classes via probability aggregation, enabling accurate inference with only one measurement shot. (*The basis space is shown only for graphical illustration. Basis states with the same label do not necessarily form a contiguous group, but may be distributed across different regions of the basis space.)

Inspired by these considerations, we propose **Y**ou **O**nly **M**easure **O**nce (**Yomo**), a simple yet effective design for shot-efficient QML. Yomo departs from the conventional Pauli expectation-value paradigm by leveraging probability aggregation of measurement outcomes, enabling accurate predictions even in the single-shot regime. Empirically, Yomo achieves competitive or superior classification accuracy compared to expectation-based baselines while requiring an order of magnitude fewer shots. This translates directly into reduced inference cost and time overhead: for a fixed budget, one can perform many more experiments, or alternatively, achieve comparable performance at a fraction of the resource usage. By bridging theoretical insights with practical implementation, Yomo demonstrates a clear pathway toward cost-efficient QML.

In summary, our contributions are as follows:

1. Introduction of Yomo framework, achieving high classification accuracy even in the single-shot inference regime.

2. Development of formal results bounding the number of measurement shots required to achieve a target error probability, proving that Yomo can surpass conventional expectation-based QML models in shot efficiency.

3. Extensive experiments on MNIST and CIFAR-10 demonstrating that Yomo consistently outperforms Vanilla QML in single-shot and few-shot regimes, validated under both noiseless simulation and simulated noise models derived from public single-qubit and two-qubit error rates of current quantum hardware.

## 2 RELATED WORKS

Research on reducing measurement overhead in quantum computing has mainly focused on *shot-efficient estimation methods* such as classical shadows (Huang et al., 2020), and on shot allocation

strategies during QML training (Phalak & Ghosh, 2023; Liang et al., 2024). More recently, Recio-Armengol et al. (2025b) introduced *single-shot QML*, while others explored *train-on-classical, deploy-on-quantum* paradigms (Duneau et al., 2024; Recio-Armengol et al., 2025a). Our work differs by addressing the inference stage, where deployment costs dominate. A more comprehensive review of related works is provided in Appendix B.

# 3 PRELIMINARY: QUANTUM MACHINE LEARNING MODELING

QML integrates the expressive power of quantum circuits with classical machine learning techniques (Cerezo et al., 2022; Huang et al., 2022; Biamonte et al., 2017; Benedetti et al., 2019). A standard workflow consists of two main components: (i) a classical feature extractor that compresses high-dimensional input data, and (ii) a quantum neural network (QNN) that processes the encoded features within a quantum state space.

**Model design.** Let the input be denoted by $x \in \mathbb{R}^d$. A classical feature extractor $f_{\theta_c}(\cdot)$, parameterized by $\theta_c$, such as a convolutional–linear network, maps it into a feature vector:

$$z = f_{\theta_c}(x) \in \mathbb{R}^{n_f}. \tag{1}$$

The extracted feature vector $z \in \mathbb{R}^{n_f}$ is encoded into the quantum circuit via angle encoding, where features are sequentially mapped to single-qubit rotations along all three axes $(R_y, R_z, R_x)$ in a cyclic fashion[1]. Specifically, each feature dimension $z_j$ parametrizes one rotation gate, assigned to a qubit following the repeating pattern $(R_y, R_z, R_x)$, creating a quantum state $|\psi(z)\rangle$ with $n_q$ qubits:

$$|\psi(z)\rangle = V(z)|0\rangle^{\otimes n_q} = \prod_{i=1}^{n_f} R_{\alpha(j)}^{i \bmod n_q}(z_i) |0\rangle^{\otimes n_q}, \quad \alpha(j) \in \{y, z, x\}, \tag{2}$$

where $\alpha(j)$ denotes the rotation axis determined by the cyclic order $(y, z, x)$, and $R_x^i$ means the $R_x$ gate is applied to the $i$-th qubit.

The encoded state is then processed by a variational circuit composed of multiple blocks of parameterized single-qubit rotations and entangling gates (e.g., CNOTs). With $n_q$ qubits and $N_b$ layers, the variational ansatz can be expressed as:

$$U(\theta) = \prod_{\ell=1}^{N_b} \left( \prod_{i=1}^{n_q-1} \text{CNOT}^{i,i+1} \prod_{j=1}^{n_q} R_y^j \left( \theta_j^{(\ell)} \right) \right). \tag{3}$$

yielding the final state $|\psi(z, \theta)\rangle = U(\theta)V(z)|0\rangle^{\otimes n_q}$.

**Pauli Expectation-based outputs.** In the conventional QML model (called Vanilla in the following of this study), class scores are obtained by measuring expectation values of observables in the computational basis. Each class $k$ is associated with a Hermitian operator $O_k$, typically chosen as a tensor product of Pauli matrices. The specific operator used in this study is provided in Appendix D. The score for class $k$ is then:

$$\mu_k = \langle \psi(z, \theta)|O_k|\psi(z, \theta)\rangle. \tag{4}$$

These scores form the logits of a softmax classifier:

$$p_k = \frac{\exp(\mu_k)}{\sum_{j=1}^K \exp(\mu_j)}, \quad \hat{y} = \arg\max_k p_k. \tag{5}$$

**Loss function.** For training the Vanilla QML model, we employ the standard cross-entropy loss to align the predicted probability distribution with the true class labels. Given logits $\mu_k$ derived from expectation values and the corresponding softmax probabilities $p_k$, the cross-entropy loss is defined as (with sample number $N_s$)

$$\mathcal{L}_{\text{CE}} = -\frac{1}{N_s} \sum_{i=1}^{N_s} y_i \log p_i, \tag{6}$$

where $y_i$ is the one-hot encoded ground-truth label and $p_i$ denotes the predicted probability for class $i$. This objective encourages the model to assign high probability to the correct class while penalizing incorrect predictions, serving as the standard baseline criterion in classification tasks.

---

[1]In general, angle encoding can be implemented using different choices or arrangements of single-qubit gates; here we present only one example.

**Measurement considerations.** Since expectation values are ensemble quantities, they cannot be extracted precisely from a single run of the quantum circuit. Instead, the circuit must be executed repeatedly, each time measuring in the computational basis, and the outcomes are averaged to approximate the expectation value. With eigenvalues bounded in $[-1, 1]$, Hoeffding's inequality shows that $N$ measurement shots reduce the estimation error at a rate $O(1/\sqrt{N})$ (more in Appendix C). Thus, reliable predictions require a large shot budget, especially when the decision margin $\Delta$ between the top-two class scores is small. As a result, expectation-based inference is both time- and resource-intensive, since quantum hardware must be reset and re-executed for every shot.

## 4 YOMO MODEL

With the motivation of reducing the measurement shot count of a QML model during inference, it is possible to investigate what kind of the design is required to make such behavior possible, construct the component, and even push the boundary to single-shot inference. Inspired by the concept of "You Only Look Once (YOLO)" (Redmon et al., 2016), we propose Yomo: **Y**ou **O**nly **M**easure **O**nce, which can achieve high testing accuracy during inference with only single-shot measurement of quantum circuit. Fig. 1 provides a comprehensive overview of Yomo.

**Model design.** Yomo shares the same construction as the expectation-based (Vanilla) QML model up to the preparation of the quantum state. An input $x \in \mathbb{R}^d$ is mapped into a feature vector $z = f_{\theta_c}(x) \in \mathbb{R}^{n_f}$ by a classical feature extractor, which is then embedded into qubits through angle encoding with cyclic rotations $(R_y, R_z, R_x)$. The encoded state is processed by a variational QNN consisting of parameterized single-qubit rotations and entangling CNOT layers, yielding the final state $|\psi(z, \theta)\rangle = U(\theta)V(z)|0\rangle^{\otimes n_q}$. Finally, unlike Vanilla QML that computes expectation values of Pauli observables, Yomo performs computational basis measurement on all qubits, producing a probability distribution over $2^{n_q}$ basis states[2]:

$$P(\phi) = \left|\langle\phi|\psi(z,\theta)\rangle\right|^2 = \langle\psi(z,\theta)|\Pi_\phi|\psi(z,\theta)\rangle, \quad \phi \in \{0,1\}^{n_q}, \ \Pi_\phi = |\phi\rangle\langle\phi|. \tag{7}$$

That is, the basis probabilities can be understood as expectation values of projection operators $\Pi_\phi$ forming the computational-basis POVM (positive operator-valued measure).

**Probability aggregation.** Since classification requires $K$ output classes, the $2^{n_q}$ computational basis states are partitioned into $K$ groups according to their index order (e.g., $0000, 0001, 0010, \dots$). Specifically, each class is assigned $\lfloor 2^{n_q}/K \rfloor$ consecutive basis states, and the remaining

$$r = 2^{n_q} - \lfloor 2^{n_q}/K \rfloor \cdot K, \qquad r \leq K$$

basis states are distributed one by one to the first $r$ classes. Let $\mathcal{S}_k$ denote the set of basis states assigned to class $k$. The aggregated probability for class $k$ is then defined as

$$p_k = \frac{1}{|\mathcal{S}_k|} \sum_{\phi \in \mathcal{S}_k} P(\phi), \qquad k = 1, \dots, K. \tag{8}$$

The final prediction is given by

$$\hat{y} = \arg\max_k p_k. \tag{9}$$

Note that with this design, although the full probability distribution is required during the training stage, at inference a single measured bitstring can be directly mapped to a class label according to the pre-defined partition of basis states. This property is the central mechanism that enables Yomo to perform accurate classification in the single-shot regime.

**Loss functions.** To stabilize training and encourage confident predictions, the Yomo loss function combines three components:

$$\mathcal{L}_{\text{yomo}} = \mathcal{L}_{\text{CE}} + \gamma \mathcal{L}_{\text{PS}} + \omega \mathcal{L}_{\text{E}}, \tag{10}$$

---

[2]At this stage, obtaining the full set of $2^{n_q}$ probabilities during training requires exact state-vector simulation. For fairness, the Vanilla baseline is also trained under exact simulation. We discuss implications of this constraint in the following sections.

where $\mathcal{L}_{\text{CE}} = -\frac{1}{N_s}\sum_{i=1}^{N_s} y_i \log p_i$ is the standard cross-entropy loss with $N_s$ samples, $y_i$ and $p_i$ denote the true label and predicted probability of the correct class for sample $i$, $\mathcal{L}_{\text{PS}}$ is a sharpening loss, and $\mathcal{L}_{\text{E}}$ enforces low-entropy distributions. The probability sharpening mechanism rewards predictions whose probability $p_i$ (corresponding to prediction $\hat{y}$) with data sample index $i$ surpasses a threshold $\tau \in (0, 1)$:

$$\mathcal{L}_{\text{PS}} = 1 - \frac{1}{|\{i\,|\,p_i > \tau\}|} \sum_{i:p_i > \tau} p_i. \tag{11}$$

This term encourages the model to push confident predictions further toward one-hot distributions. As one can observe, if there are no prediction $p_i$ larger than $\tau$, then $\mathcal{L}_{\text{PS}}$ is 1, and $\mathcal{L}_{\text{PS}}$ will be close to 0 if $p_i > \tau$ and $p_i \to 1$. In addition, we introduce an entropy regularization term

$$\mathcal{L}_{\text{E}} = -\frac{1}{N_s} \sum_{i=1}^{N_s} p_i \log p_i, \tag{12}$$

which penalizes flat probability distributions and promotes sharper decision boundaries. The hyper-parameters $\gamma$ and $\omega$ control the relative strengths of sharpening and entropy regularization.

## 5 THEORETICAL RESULTS

We summarize the main theoretical findings supporting the proposed Yomo framework. Complete derivations and proofs are provided in Appendix C.

**Theorem 5.1** (Shot requirement of Yomo). *Let $p > \frac{1}{2}$ denote the probability that a single-shot measurement yields the correct class in the trained Yomo model. To achieve $\Pr(\text{incorrect}) \le \delta$, with $\Pr(\text{incorrect}) \le \exp\left(-2N(p-\frac{1}{2})^2\right)$, a sufficient shot budget $N_{yo}$ is:*

$$N_{yo} \ge \frac{\ln(1/\delta)}{2\left(p-\frac{1}{2}\right)^2} \tag{13}$$

*Here $\Pr(\text{incorrect})$ denotes the misclassification probability under finite-shot sampling, and $\delta \in (0, 1)$ is the target error tolerance. (For even N, assume adversarial tie-break; random ties change only constants.)*

**Theorem 5.2** (Shot requirement of Vanilla QML). *Let $\Delta$ be the minimum margin between the top-two class scores in the infinite-shot limit, $L$ be the Lipschitz constant of the score–expectation mapping, and $K$ the number of classes. To achieve $\Pr(\text{incorrect}) \le \delta$, with $\Pr(\text{incorrect}) \le 2K \exp\left(-2N\left(\frac{\Delta}{4L}\right)^2\right)$, a sufficient shot budget $N_{va}$ is:*

$$N_{va} \ge \frac{8L^2}{\Delta^2} \ln\frac{2K}{\delta} \tag{14}$$

**Theorem 5.3** (Condition for fewer shots at fixed $\delta$). *Fix $\delta$. If the trained Yomo model achieves*

$$p \ge \frac{1}{2} + \frac{\Delta}{4L} \sqrt{\frac{\ln(1/\delta)}{\ln(2K/\delta)}}, \tag{15}$$

*then Yomo requires fewer measurement shots than Vanilla QML to reach the same target error probability $\delta$.*

**Theorem 5.4** (Condition for smaller $\delta$ at fixed $N$). *Fix $N$. If the trained Yomo model achieves*

$$p \ge \frac{1}{2} + \sqrt{\left(\frac{\Delta}{4L}\right)^2 - \frac{\ln(2K)}{2N}}, \tag{16}$$

*then Yomo attains a smaller incorrect probability $\delta$ than Vanilla QML with the same shot budget $N$.*

**Theorem 5.5** (Single-shot condition). *In the $N = 1$ regime, if the trained Yomo model achieves*

$$p \ge 1 - 2K \exp\left(-\frac{\Delta^2}{8L^2}\right), \tag{17}$$

*then its incorrect probability $\delta$ is guaranteed to be smaller than that of Vanilla QML.*

The above results reveal that Vanilla is fundamentally disadvantaged in two ways. First, its bound carries a multiplicative $2K$ factor from the union bound over $K$ classes, which directly inflates the shot requirement. Second, its dependence on the top-two score margin $\Delta$ means that if $\Delta$ shrinks, often exponentially with qubit count or under noise (McClean et al., 2018), then the required $N_{va}$ in Eq. 14 grows exponentially as well. In contrast, Yomo's requirement in Eq. 13 depends only on $p - \frac{1}{2}$, determined by training, which can remain stable as qubit count increases, enabling Yomo to sustain low-shot performance even in high-dimensional regimes where Vanilla QML becomes impractical during inference.

## 6   EXPERIMENTS

We validate the effectiveness of Yomo by comparing it against Vanilla QML models. Hyperparameter and training settings for all experiments are provided in Appendix D. Reported results are averaged over 5 runs with different random seeds to ensure robustness.

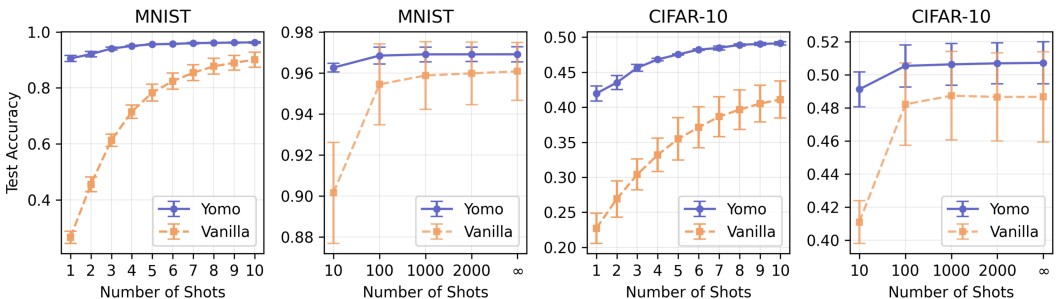

Figure 2: Comparison of test accuracy between Yomo (blue, solid line) and Vanilla (orange, dashed line) models across different shot budgets on MNIST (left two panels) and CIFAR-10 (right two panels). Both models use $n_q = 4$ qubits and $N_b = 5$ QNN blocks. For Yomo, the threshold parameter is fixed at $\tau = 0.6$.

**Single-shot behavior of Yomo.**   We first evaluate Yomo and Vanilla on MNIST and CIFAR-10 classification tasks under varying shot budgets. With the network structure fixed as described in Sec. 4, these experiments directly assess the contribution of Yomo's probability-based output and loss design. As shown in Fig. 2, under the single-shot regime (shots = 1) on MNIST, Vanilla achieves only 26.59% test accuracy, whereas Yomo attains 90.52%. Increasing the number of shots substantially improves Vanilla: at shots = 10, its accuracy reaches 90.08%, comparable to Yomo's single-shot performance. This indicates that Yomo can achieve the same level of accuracy with roughly $10\times$ fewer shots. Notably, Yomo also continues to improve as the shot budget increases. In Fig. 2, we further extend the evaluation up to shots $\rightarrow \infty$, corresponding to exact state-vector simulation. Across the full shot regime, Yomo consistently outperforms Vanilla. Similar trends are observed on the more challenging CIFAR-10 task, where Yomo maintains its advantage in both the low- and high-shot settings.

**Effects of qubit count.**   As discussed in Sec. 5, the required number of shots for Vanilla grows with decreasing top-two score margin $\Delta$, which typically shrinks as the system size increases. The sufficient budget scales as $N_{va} \geq O(1/\Delta^2)$, implying severe shot requirements for larger $n_q$. In contrast, Yomo is not constrained by this dependence. To verify this, we compare both models with increasing qubit counts, $n_q \in \{4, 6, 8, 10, 12\}$, while fixing $N_b = 5$. Results in Fig. 3(a,b) clearly show that Vanilla suffers significant performance degradation as $n_q$ increases. Although larger qubit counts introduce more trainable parameters, Vanilla would require deeper circuits (larger $N_b$) to maintain expressivity. On the other hand, larger $n_q$ can also allow shallower input encodings for a fixed number of features, which is advantageous in the noisy intermediate-scale quantum (NISQ) (Preskill, 2018) regime. Yomo, however, shows no comparable performance decay as $n_q$ grows, consistent with our theoretical findings in Sec. 5.

**Is threshold $\tau$ important?** During training, Yomo employs a probability sharpening mechanism (Eq. 11). Intuitively, setting $\tau$ too low may amplify incorrect predictions early in training, while setting it too high primarily enhances already confident predictions, providing limited benefit. Hence, an intermediate threshold is expected to be most effective. Figure 3(c) shows test accuracy across different $\tau$ values and shot budgets. While results for shots $> 10$ are relatively insensitive to $\tau$, in the single-shot regime the accuracy peaks at $\tau = 0.6$, confirming this moderate choice as optimal.

**Probability sharpening mechanism.** Figures 3(d,e) compare training dynamics of Yomo with and without the sharpening loss $\mathcal{L}_{\mathrm{PS}}$ (Eq. 11). The inclusion of $\mathcal{L}_{\mathrm{PS}}$ clearly improves single-shot test accuracy, as shown in Fig. 3(e), validating the effectiveness of this mechanism in guiding the model toward more confident and accurate predictions.

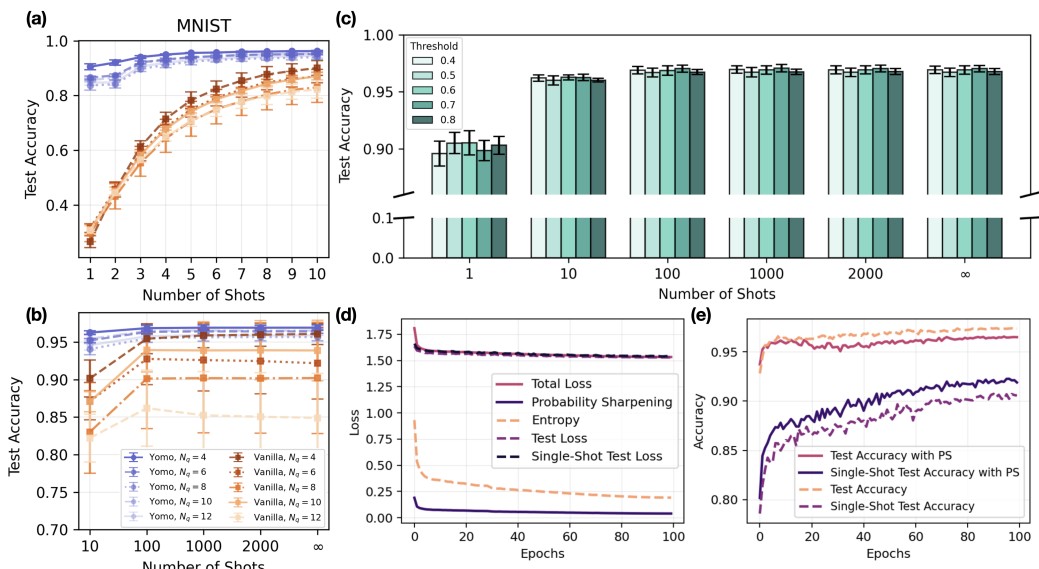

Figure 3: Extended evaluation of Yomo and Vanilla models. (a, b) Effect of qubit count $n_q \in 4, 6, 8, 10, 12$ on MNIST accuracy. (c) Sensitivity to threshold $\tau \in 0.4, 0.5, 0.6, 0.7, 0.8$. While performance is similar for shots $> 10$, single-shot accuracy peaks at $\tau = 0.6$. (d) Training loss decomposition of Yomo into total loss, probability sharpening (PS) loss, entropy loss, and test losses. (e) Test accuracy trajectories with and without the sharpening loss $\mathcal{L}_{\mathrm{PS}}$ ($\tau = 0.6$).

**Noisy simulation with deeper QNN.** Since Yomo targets shot-efficient inference, it is crucial to test its reliability under realistic NISQ noise. To this end, we approximate hardware noise using depolarizing channels applied to both single-qubit (1Q) and two-qubit (2Q) operations (details provided in Appendix E). Noise model approximations for different quantum hardware platforms are constructed by mapping publicly reported 1Q and 2Q error rates to depolarizing error probabilities, as summarized in Table 1. Figs. 4 and 5 present noisy simulation results for Yomo and Vanilla on MNIST and CIFAR-10, respectively, evaluated across shot budgets $N_{\mathrm{shot}} \in \{1, 100, \infty\}$ and different numbers of QNN blocks. The simulated hardware noise model includes Quantinuum H1-1, IBM_Pittsburgh, Google Willow, and IonQ Forte. Among these, Quantinuum H1-1 exhibits performance closest to the noiseless baseline, followed by IBM_Pittsburgh, Google Willow, and IonQ Forte. This ordering mirrors their reported 2Q error rates, indicating that as circuit depth increases, the 2Q error rate becomes the dominant factor governing overall model accuracy.

Figs. 4 and 5 further reveal a clear contrast between Vanilla and Yomo in terms of depth–performance behavior under noisy conditions. For Vanilla, at sufficiently large shot budgets ($N_{\mathrm{shot}} = 100, \infty$), test accuracy initially increases with the number of QNN blocks, reaching a sweet spot around 10–15 blocks before degrading as noise accumulates. This indicates that, when enough measurement precision is available, additional expressiveness from deeper circuits can momentarily outweigh the effects of noise. In contrast, Yomo already achieves strong accuracy at very low depth ($N_b = 5$),

Table 1: Depolarizing Noise Level Reference (IBM Quantum, 2025; Google Quantum AI, 2024; Quantinuum Systems, 2025; IonQ, 2025). We note that the IBM Quantum platform's error rates fluctuate over time. The data presented was recorded on August 28, 2025. For IBM, the 1Q error rates reported are median values, while the 2Q error rates are average values. For all other providers, the reported error rates are average values.

| Device | 1-Qubit Error Rate | 2-Qubit Error Rate | Approx. Depolarizing $p_1/p_2$ |
|---|---|---|---|
| IBM_Pittsburgh | 0.0202% | 0.169% | $p_1 \sim 2.02 \times 10^{-4}$, $p_2 \sim 1.69 \times 10^{-3}$ |
| Google Willow | 0.035% | 0.33% | $p_1 \sim 3.5 \times 10^{-4}$, $p_2 \sim 3.3 \times 10^{-3}$ |
| Quantinuum H1-1 | 0.0018% | 0.097% | $p_1 \sim 1.8 \times 10^{-5}$, $p_2 \sim 9.7 \times 10^{-4}$ |
| IonQ Forte | 0.02% | 0.4% | $p_1 \sim 2 \times 10^{-4}$, $p_2 \sim 4 \times 10^{-3}$ |

leaving little room for further improvement. As a result, deeper circuits do not provide additional benefit, and performance decreases monotonically due to noise accumulation. This distinction highlights a fundamental difference: while Vanilla relies on deeper circuits and larger shot budgets to exploit expressiveness, Yomo is suitable for low-depth, shot-efficient inference. Moreover, Yomo remains robust on noisy settings, with performance in some hardware configurations (e.g., Quantinuum H1-1) closely tracking the noiseless baseline. Even in the single-shot regime, Yomo matches the accuracy of Vanilla models that require orders of magnitude more measurements, representing its practical advantage in both runtime and hardware cost.

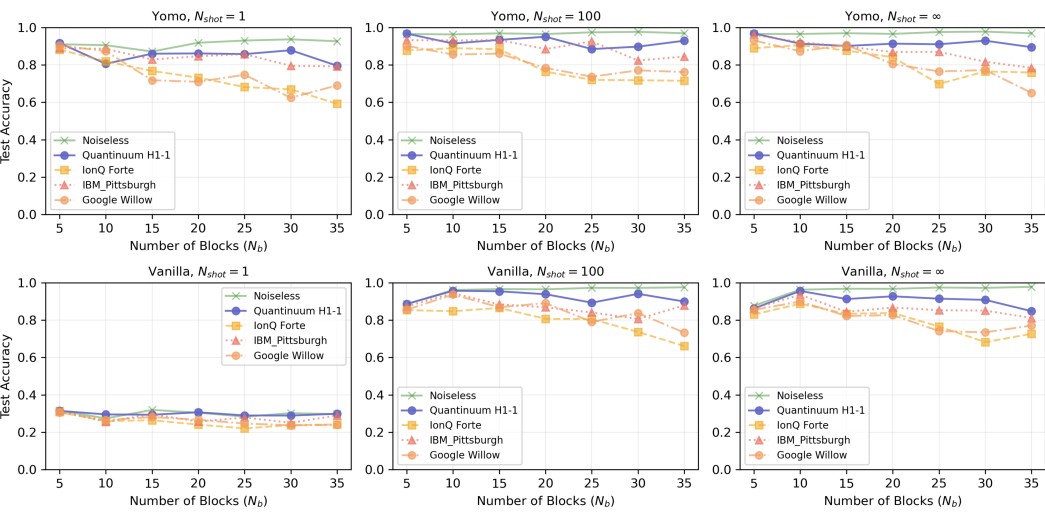

Figure 4: Comparison of Yomo and Vanilla QML models on MNIST under different hardware noise settings and shot budgets. Top row: Yomo with $N_{\text{shot}} \in \{1, 100, \infty\}$. Bottom row: Vanilla with $N_{\text{shot}} \in \{1, 100, \infty\}$. Each curve shows test accuracy as a function of the number of circuit blocks $N_b$ under noiseless simulation and depolarizing noise models parameterized by hardware benchmarks from Quantinuum H1-1, IonQ Forte, IBM_Pittsburgh, and Google Willow.

## 7 DISCUSSION AND CONCLUSION

Our experiments demonstrate that Yomo achieves competitive or even superior performance with dramatically fewer measurement shots compared to Vanilla QML models. In some cases, Yomo attains high test accuracy with only a single shot. This has immediate implications for the practical use of quantum hardware. Since providers typically charge in proportion to the number of shots or runtime, reducing the required shots translates directly into lower usage costs. Conversely, under a fixed budget, users could conduct significantly more experiments or obtain higher-quality results. In this sense, Yomo contributes to lowering the economic barrier of adopting quantum technologies in both academic and industrial settings.

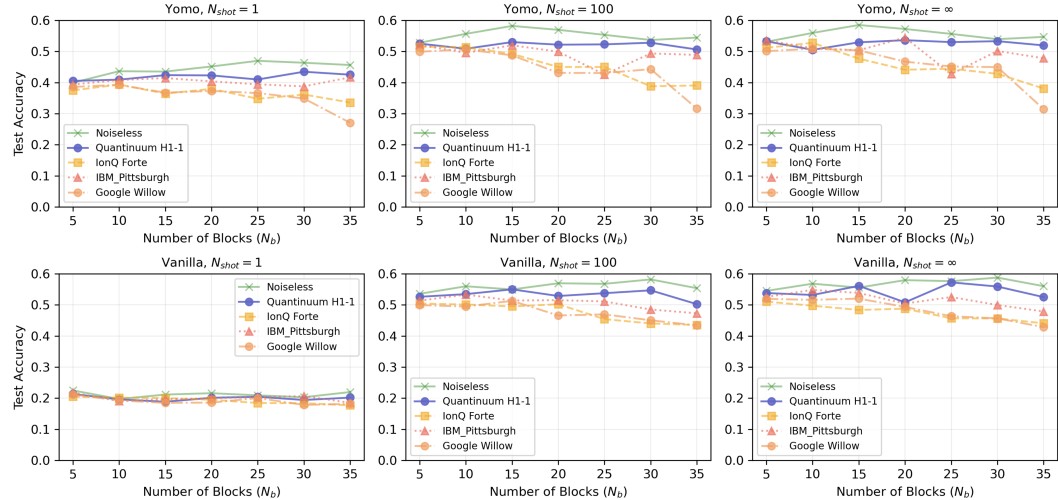

Figure 5: Comparison of Yomo and Vanilla QML models on CIFAR-10 under different hardware noise settings and shot budgets. Top row: Yomo with $N_{\text{shot}} \in \{1, 100, \infty\}$. Bottom row: Vanilla with $N_{\text{shot}} \in \{1, 100, \infty\}$. Each curve shows test accuracy as a function of the number of circuit blocks $N_b$ under noiseless simulation and depolarizing noise models parameterized by hardware benchmarks from Quantinuum H1-1, IonQ Forte, IBM_Pittsburgh, and Google Willow.

It is important to emphasize that Yomo is not intended to be trained directly on quantum hardware. Instead, its design is particularly well-suited to the setting where training is performed using classical simulation of quantum states, while deployment takes place on quantum devices. This separation leverages the flexibility of classical training environments, avoiding the substantial shot cost and noise challenges of on-hardware optimization. In the inference stage, however, Yomo's single-shot capability enables efficient execution on real quantum processors. Notably, in the intermediate qubit regime (e.g., 25–35 qubits), quantum inference with Yomo may even surpass classical simulation in runtime, as suggested by (Chatterjee et al., 2025), due to the intrinsic efficiency of single-shot execution. A systematic investigation of this crossover point, which we leave for future work, could provide valuable guidance for determining when quantum inference becomes advantageous in practice.

Because training remains more efficient and practical on classical hardware in small qubit size, to scale up, an important future direction is to explore advanced classical methods for simulating QNN outputs. For example, the *train-on-classical, deploy-on-quantum* paradigm (Duneau et al., 2024; Recio-Armengol et al., 2025a) highlights the possibility of scalable classical training pipelines. Integrating such methods with shot-efficient inference schemes like Yomo may further reduce the total cost of deploying QML models.

Our inference-stage evaluation explicitly accounts for realistic practical constraints, finite measurement shots and noisy environment. We employed error models parameterized by 1Q and 2Q depolarizing noise derived from publicly available error rates. While these models cannot capture all device-specific imperfections, they offer a reasonable proxy for the effects of hardware noise. Importantly, our results show that Yomo maintains robust single-shot behavior even under these noise conditions. We note, however, that real devices such as IBM_Pittsburgh or Google Willow have limited qubit connectivity, which would require additional SWAP gates compared to the fully connected ion-trap architectures of IonQ and Quantinuum. This connectivity overhead may further degrade performance in practice, suggesting that Yomo's advantage could be even more pronounced on hardware with higher connectivity.

By enabling accurate single-shot inference, Yomo reduces costs of deploying QML, thereby making quantum models more accessible. Looking forward, combining Yomo with advances in classical simulation techniques, scaling analyses of the quantum-classical crossover regime, and device-aware optimizations will further advance the feasibility of practical QML deployment.

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

## A  MOTIVATION: THE IMPORTANCE OF SHOT EFFICIENCY FOR INFERENCE

Most prior research on QML has focused on improving the training phase, such as optimizing gradients or reducing the number of circuit evaluations required during parameter updates. However, scaling studies in classical machine learning have shown that, once a model is deployed at scale, the dominant cost often shifts from training to inference (Sardana et al., 2023). The same trend is expected for QML, such that when quantum hardware becomes routinely accessible, inference will constitute the primary driver of both computational and monetary cost. In such a setting, shot efficiency during inference becomes essential. From a practical perspective, achieving competitive performance with a fraction of the measurement shots would offer a decisive advantage. As illustrated in Fig. 6, if the target ac-

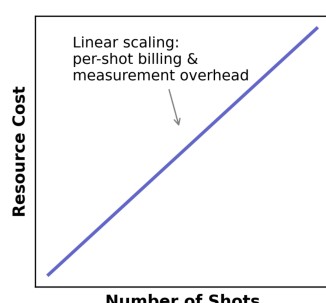

Figure 6: Resource cost scales linearly with the number of measurement shots due to both per-shot billing and measurement overhead.

curacy can be reached using only a handful of shots, the savings in hardware usage grow proportionally with the reduction in shots[3]. This means that, under a fixed quantum computing budget, a researcher or practitioner could run many more experiments, accelerating scientific progress and enabling broader adoption in industrial applications. Conversely, for a fixed workload, the overall inference cost could be reduced by orders of magnitude. In both scenarios, shot-efficient inference directly lowers the barrier to practical deployment of QML.

## B  EXPANDED RELATED WORKS

**Shot-efficient estimation methods.**  A prominent line of work focuses on minimizing the number of circuit executions required to extract useful information. For instance, the classical shadow framework (Huang et al., 2020) reuses measurement data to predict many observables simultaneously. While highly effective for general quantum state tomography, these approaches are not specifically designed for QML tasks, where the objective is to train and deploy predictive models efficiently.

**Shot-efficient QML.**  Within the QML literature, shot optimization has primarily been investigated in the context of *training*. Several works propose adaptive or distribution-aware strategies to allocate shots during training epochs (Phalak & Ghosh, 2023; Liang et al., 2024), thereby accelerating convergence while preserving accuracy. Although these methods demonstrate that judicious shot allocation can substantially reduce training cost, the models still rely on expectation-value outputs at inference time. As such, they do not directly address the cost of deployment, where inference calls may dominate the lifecycle usage of a machine learning model. More recently, Recio-Armengol et al. (2025b) introduced the concept of *single-shot QML*, providing a theoretical characterization of when a QML model can achieve reliable predictions with only a single measurement. Their work highlights both the promise and the difficulty of realizing single-shot models in practice. While the potential cost savings are significant, training such models directly is shown to be challenging. Despite the importance of this direction, there has been limited follow-up work, largely due to the absence of a concrete architectural design or implementation pathway.

**Training on classical hardware.**  Complementary to these lines of research, other efforts have investigated hybrid training settings such as *train-on-classical, deploy-on-quantum*, where models are trained using classical simulations and then deployed on real quantum hardware (Duneau et al., 2024; Recio-Armengol et al., 2025a). These works primarily address the training bottleneck imposed by scarce quantum resources and the challenges of gradient evaluation. In contrast, our work focuses on the inference stage. Nevertheless, the *train-on-classical* paradigm offers a promising pathway for scaling up the models proposed here in future work.

---

[3]Quantum hardware providers typically charge in proportion to the number of measurement shots or to execution time, which itself scales linearly with shots. For example, the IBM Quantum Platform pricing page (https://www.ibm.com/quantum/products) and IonQ on AWS Braket pricing page (https://aws.amazon.com/braket/pricing/).

## C  THEORETICAL RESULTS ON INFERENCE SHOT REQUIREMENT

In the main paper, we stated several theoretical guarantees on the shot requirements for inference in both expectation-based (Vanilla) QML and probability-aggregation (Yomo) QML. For completeness, we provide the detailed proofs here.

**Vanilla QML.**  Expectation-based QML models produce predictions by computing class scores $s_c = g_c(\boldsymbol{\mu})$, where $\boldsymbol{\mu} = (\mu_1, \ldots, \mu_K)$ are expectation values $\mu_c = \langle O_c \rangle$ of class observables $\{O_c\}$. The predicted label is then given by $\arg\max_c \text{softmax}(s_c)$. We adopt the following assumptions:

- **Bounded outcomes.** Each single-shot outcome used to estimate $\mu_c$ lies in $[-1, 1]$ (e.g., Pauli eigenvalues $\pm 1$).
- **Lipschitz scores.** The score map $s = g(\boldsymbol{\mu})$ is $L$-Lipschitz under $\|\cdot\|_\infty$, i.e.,

$$|s_c(\hat{\boldsymbol{\mu}}) - s_c(\boldsymbol{\mu})| \le L\|\hat{\boldsymbol{\mu}} - \boldsymbol{\mu}\|_\infty.$$

  For linear logits, one has $L = \|W\|_\infty$.
- **Margin.** Let $s_{(1)} > s_{(2)}$ denote the top-two true scores and define the margin $\Delta := s_{(1)} - s_{(2)} > 0$.

**Lemma C.1** (Concentration of expectations). *For each class $c$, with $N$ i.i.d. measurement shots and estimator $\hat{\mu}_c$,*

$$\Pr\big(|\hat{\mu}_c - \mu_c| \ge \varepsilon\big) \le 2\exp(-2N\varepsilon^2). \tag{18}$$

**Proposition C.2** (Argmax stability under margin). *If $\|\hat{\boldsymbol{\mu}} - \boldsymbol{\mu}\|_\infty \le \Delta/(4L)$, then the predicted class is preserved, i.e., $\arg\max_c \hat{s}_c = \arg\max_c s_c$.*

*Proof sketch.*  By Lipschitz continuity,

$$\max_c |\hat{s}_c - s_c| \le L\|\hat{\boldsymbol{\mu}} - \boldsymbol{\mu}\|_\infty \le \Delta/4.$$

Thus,

$$\hat{s}_{(1)} - \hat{s}_{(2)} \ge \big(s_{(1)} - \tfrac{\Delta}{4}\big) - \big(s_{(2)} + \tfrac{\Delta}{4}\big) = \tfrac{\Delta}{2} > 0,$$

ensuring that the argmax is unchanged. $\qquad\square$

**Corollary C.2.1** (Decision tail bound). *Using a union bound over $K$ classes and Lemma C.1,*

$$\Pr(\textit{incorrect}) \le \Pr\Big(\max_c |\hat{\mu}_c - \mu_c| > \tfrac{\Delta}{4L}\Big) \le 2K\exp\Big(-2N\big(\tfrac{\Delta}{4L}\big)^2\Big). \tag{19}$$

*Equivalently, to guarantee $\Pr(\textit{incorrect}) \le \delta$, it suffices to take*

$$N \ge \frac{8L^2}{\Delta^2}\ln\frac{2K}{\delta}. \tag{20}$$

Hence the decision error probability decays in $N$, but the required shot budget scales with $\frac{L^2}{\Delta^2}$. As system size grows, margins $\Delta$ often shrink while Lipschitz constants $L$ increase, inflating the required $N$ and making Vanilla QML inference shot-inefficient.

**Yomo QML.**  In Yomo, inference is based on direct measurement outcomes. Let $p \in (0, 1)$ denote the probability that a single shot yields the correct class label (i.e., the true class receives the majority of aggregated probability). With $N$ i.i.d. shots and majority vote, the number of correct votes follows $S_N \sim \text{Binomial}(N, p)$.

**Proposition C.3** (Binomial tail bound for majority vote). *For odd $N$ (ties can be handled analogously), the error probability is*

$$\Pr(\textit{incorrect}) = \Pr\big(S_N < \tfrac{N}{2}\big) = \sum_{k=0}^{\lceil N/2 \rceil - 1} \binom{N}{k} p^k (1-p)^{N-k} \le \exp\big(-2N(p - \tfrac{1}{2})^2\big). \tag{21}$$

*Thus, to ensure $\Pr(\textit{incorrect}) \le \delta$, it suffices to take*

$$N \ge \frac{\ln(1/\delta)}{2(p - \tfrac{1}{2})^2}. \tag{22}$$

In practice, training typically yields $p \geq 0.85$–$0.95$, in which case $N = 3$–$5$ already achieves $> 99\%$ reliability. This theoretical guarantee aligns with our empirical observations and explains the rapid accuracy gains observed when moving from single-shot to small-shot inference in Yomo.

**Theorem C.4** (Condition for fewer shots at fixed $\delta$). *Fix a target error level $\delta \in (0, 1)$. If the trained Yomo model satisfies*

$$p \geq \frac{1}{2} + \frac{\Delta}{4L} \sqrt{\frac{\ln(1/\delta)}{\ln(2K/\delta)}},$$

*then Yomo requires no more (and strictly fewer whenever the inequality is strict) measurement shots than Vanilla to reach error at most $\delta$.*

*Proof.* From the bounds above, $N_{\mathrm{yo}}(\delta) \geq \frac{\ln(1/\delta)}{2(p-\frac{1}{2})^2}$ and $N_{\mathrm{va}}(\delta) \geq \frac{8L^2}{\Delta^2} \ln \frac{2K}{\delta}$. Requiring $N_{\mathrm{yo}}(\delta) \leq N_{\mathrm{va}}(\delta)$ gives $\frac{\ln(1/\delta)}{2(p-\frac{1}{2})^2} \leq \frac{8L^2}{\Delta^2} \ln \frac{2K}{\delta}$, equivalently $(p - \frac{1}{2})^2 \geq \frac{\Delta^2}{16L^2} \frac{\ln(1/\delta)}{\ln(2K/\delta)}$. Taking square roots proves the claim. $\square$

**Theorem C.5** (Condition for smaller $\delta$ at fixed $N$). *Fix a shot budget $N \in \mathbb{N}$. If the trained Yomo model satisfies*

$$p \geq \frac{1}{2} + \sqrt{\left(\frac{\Delta}{4L}\right)^2 - \frac{\ln(2K)}{2N}},$$

*then Yomo attains a smaller error probability than Vanilla with the same $N$ shots.*

*Proof.* We require $\delta_{\mathrm{yo}} \leq \delta_{\mathrm{va}}$, i.e. $e^{-2N(p-\frac{1}{2})^2} \leq 2K e^{-2N(\Delta/4L)^2}$. Taking logs yields $-2N(p - \frac{1}{2})^2 \leq \ln(2K) - 2N(\Delta/4L)^2$ and hence $(p - \frac{1}{2})^2 \geq (\Delta/4L)^2 - \frac{\ln(2K)}{2N}$. Taking square roots gives the result (the condition is non-vacuous when the term under the square root is nonnegative). $\square$

**Theorem C.6** (Single-shot condition). *In the $N = 1$ regime, if the trained Yomo model satisfies*

$$p \geq 1 - 2K \exp\left(-\frac{\Delta^2}{8L^2}\right),$$

*then Yomo's error probability is no larger than Vanilla's.*

*Proof.* For $N = 1$, Yomo's error is exact: $\delta_{\mathrm{yo}} = 1 - p$. Vanilla's bound gives $\delta_{\mathrm{va}} \leq 2K \exp\left[-2(\Delta/4L)^2\right] = 2K \exp\left[-\frac{\Delta^2}{8L^2}\right]$. Requiring $1 - p \leq \delta_{\mathrm{va}}$ yields the stated inequality. $\square$

**Concluding remark.** Taken together, these results show a clear separation between expectation-based (Vanilla) QML and probability-aggregation (Yomo) QML in terms of inference shot complexity. For Vanilla, the required number of shots scales inversely with the square of the classification margin $\Delta$ and grows with the Lipschitz constant $L$, both of which typically worsen with circuit size and noise. In contrast, Yomo's requirement depends only on the single-shot correctness probability $p$, which is directly controlled by training. As a result, once training produces $p$ moderately above $1/2$, Yomo achieves reliable inference with only a handful of shots, often orders of magnitude fewer than Vanilla. This theoretical advantage explains and complements the empirical findings reported in the main text.

Table 2: Comparison of shot complexity between Vanilla (expectation-based) QML and Yomo (probability-aggregation) QML. Bounds are up to constant factors and logarithmic terms.

|  | **Vanilla QML** | **Yomo QML** |
|---|---|---|
| Error bound | $\delta_{\mathrm{va}} \leq 2K \exp\left(-2N(\frac{\Delta}{4L})^2\right)$ | $\delta_{\mathrm{yo}} \leq \exp\left(-2N(p - \frac{1}{2})^2\right)$ |
| Shots for target $\delta$ | $N \geq \frac{8L^2}{\Delta^2} \ln \frac{2K}{\delta}$ | $N \geq \frac{\ln(1/\delta)}{2(p-\frac{1}{2})^2}$ |
| Single-shot error | $\delta_{\mathrm{va}} \leq 2K e^{-\Delta^2/(8L^2)}$ | $\delta_{\mathrm{yo}} = 1 - p$ |

# D  HYPERPARAMETER AND TRAINING SETTINGS IN EXPERIMENTS

**Software and Hardware.**  All experiments were performed on a system equipped with 8 NVIDIA A100 GPUs. The implementation was based on the TorchQuantum framework (Wang et al., 2022). The code for this study will be released publicly on GitHub in the coming months.

**Optimizer and Learning Rate.**  For classification tasks (MNIST and CIFAR-10), we used the Adam optimizer with a learning rate of $5 \times 10^{-3}$ for MNIST and $1 \times 10^{-3}$ for CIFAR-10.

**Batch Size and Epochs.**  Batch size was set to 128 for MNIST classification tasks and 64 for CIFAR10 classification tasks. All models were trained for 100 epochs.

**Loss Coefficients.**  The weighting coefficients $\gamma$ and $\omega$ in the total loss function (Eq. 10) are both fixed to 0.05 throughout our experiments.

**Pauli Observables in Vanilla QML.**  In Vanilla QML, each class is associated with a Hermitian observable constructed from tensor products of Pauli operators. Following prior works, we select a fixed set of 10 observables as the measurement target. For the case of $n_q = 4$ qubits and 10-class, these are

$$\{\, ZIII,\ IZII,\ IIZI,\ IIIZ,\ ZZII,\ ZIZI,\ IZZI,\ IIZZ,\ YIYI,\ IYIY \,\},$$

where $X, Y, Z$ denote Pauli matrices and $I$ is the identity. For larger numbers of qubits ($n_q > 4$), the observables are extended by appending identity operators to the right, ensuring that they act non-trivially only on the first four qubits. This construction provides a consistent set for classification tasks, while maintaining scalability across different circuit widths.

**Training procedure.**  Yomo models are trained entirely on classical simulators of quantum states, where exact probability distributions $P(\phi)$ can be computed. For fairness, the Vanilla baseline is likewise evaluated using exact state-vector simulation. The trainable parameters consist of both the classical feature extractor parameters $\theta_c$ and the quantum circuit parameters $\theta$. Given the aggregated class probabilities $\{p_k\}_{k=1}^K$ defined in Eq. 8, the training objective is the total loss $\mathcal{L}_{\text{yomo}}$ (Eq. 10). Optimization proceeds by computing gradients with respect to $(\theta_c, \theta)$. Formally, for a parameterized hybrid model

$$z = f_{\theta_c}(x), \qquad |\psi(z,\theta)\rangle = U(\theta)\, V(z)\, |0^{\otimes n_q}\rangle,$$

the aggregated class probability for class $k$ is

$$p_k(x; \theta_c, \theta) = \frac{1}{|\mathcal{S}_k|} \sum_{\phi \in \mathcal{S}_k} \left| \langle \phi | \psi(z,\theta) \rangle \right|^2 .$$

Gradients with respect to $\theta_c$ are obtained via backpropagation, whereas gradients with respect to the quantum parameters $\theta$ are, in principle, evaluated using the parameter-shift rule (Schuld et al., 2019). In our simulations, however, both $\theta_c$ and $\theta$ are updated using PyTorch's automatic differentiation engine (`autograd`). The parameter updates follow the standard form

$$(\theta_c^{(t+1)}, \theta^{(t+1)}) = (\theta_c^{(t)}, \theta^{(t)}) - \eta \, \nabla_{(\theta_c, \theta)} \mathcal{L}_{\text{yomo}}(\theta_c^{(t)}, \theta^{(t)}), \tag{23}$$

where $\eta$ is the learning rate and $\nabla_{(\theta_c, \theta)}$ denotes the joint gradient. In our experiments we employed the Adam optimizer for stability.

We emphasize that this training is performed entirely on classical simulators, avoiding the prohibitive shot cost of gradient estimation on quantum devices. The trained parameters $(\theta_c^\star, \theta^\star)$ are then deployed for inference, where Yomo's shot-efficient prediction mechanism eliminates the need to reconstruct Pauli expectation values.

# E  NOISY SIMULATION WITH DEPOLARIZING ERROR

The depolarizing error has been widely used as a baseline noise model in studies of NISQ algorithms, including QML and VQAs (Preskill, 2018; Bravyi et al., 2020; Stilck França & Garcia-Patron, 2021). In particular, several works simulate device behavior by mapping reported hardware

gate error rates directly to depolarizing error probabilities (Wang et al., 2021; Saib et al., 2021; Wood, 2020). While real hardware noise is typically biased and correlated, the depolarizing approximation is a first-order approximation of the noise, capturing the dominant effect of error rates on algorithmic performance.

The *depolarizing channel* is a quantum noise process that modifies any state towards a maximally mixed state. For any $d$-dimensional system ($d$ referred as the number of qubits), the quantum system subjected to depolarizing noise is defined as:

$$\mathcal{E}_{\text{dep}}^{(d)}(\rho) \ = \ (1-p)\rho \ + \ \tfrac{p}{d}\,I_d, \tag{24}$$

where $I_d$ is the $d$-dimensional identity operator (Nielsen & Chuang, 2010). While this could be modeled with extra control qubits, the practical implementation follows an equivalent definition which applied over time statistically matches Eq. 24. For a 1-qubit system, given any arbitrary quantum state $\rho$, it holds:

$$\frac{I}{2} = \frac{\rho + X\rho X + Y\rho Y + Z\rho Z}{4} \tag{25}$$

where $X, Y, Z$ are the Pauli operators. Therefore, substituting Eq. 25 to Eq. 24 and reparametrizing $p$, we can write the depolarizing channel as:

$$\mathcal{E}_{\text{dep}}^{(1)}(\rho) \ = \ (1-p)\rho \ + \ \tfrac{p}{3}\big(X\rho X + Y\rho Y + Z\rho Z\big), \tag{26}$$

Following the same logic, we can write the 2-qubit depolarizing channel as:

$$\mathcal{E}_{\text{dep}}^{(2)}(\rho) \ = \ (1-p)\rho \ + \ \tfrac{p}{15} \sum_{P \in \mathcal{P}_2 \setminus \{II\}} P\rho P, \tag{27}$$

where $\mathcal{P}_2 = \{I, X, Y, Z\}^{\otimes 2}$ denotes the two-qubit Pauli group and $II$ is excluded from the summation. This definition generalizes naturally to $n$ qubits: the channel acts by leaving the state unchanged with probability $(1-p)$, while with probability $p$ it applies one of the $4^n - 1$ non-identity Pauli operators uniformly at random.

In practice, depolarizing noise is typically applied after each gate, with separate parameters $p_{1Q}$ and $p_{2Q}$ for 1-qubit and 2-qubit operations, respectively. These parameters are often chosen to match the error rates reported by quantum hardware providers. For example, if a device specifies a 2-qubit gate error rate of $1.0 \times 10^{-3}$, one may simulate it by applying a 2-qubit depolarizing channel with $p_{2Q} = 10^{-3}$ after each entangling gate.The quantum circuits of our work comprises only of 1 and 2-qubit gates ($RY$, $RZ$, $RX$ and $CNOT$), therefore we modify our quantum circuits following Eqs. 24 and 27 (applying probabilistically the depolarizing Pauli operations) with the probabilities $p_1$ and $p_2$ (see Table 1 in the main text) that approximate the quantum devices 1 and 2-qubit errors.

## F    Quantum Methods in Different Training/Deployment Schemes

The design of Yomo is most naturally suited to the *train-on-classical, deploy-on-quantum* paradigm, where training can be performed efficiently on simulators and inference leverages quantum hardware with shot-efficient measurement. To place this in context, we summarize and contrast different training and deployment schemes that have been explored in the literature.

**Training & deployment on classical.**    This category corresponds to so-called *quantum-inspired* methods. Here, both training and inference are performed entirely on classical hardware, while the model architecture is motivated by quantum principles such as tensor networks, parameterized unitary evolutions, or measurement-based output mechanisms (Koike-Akino et al., 2025; Huynh et al., 2023). In the example of QuanTA (Chen et al., 2024), which introduces theoretical constructs inspired by quantum states but evaluates them using classical simulation. These methods are advantageous when quantum hardware is unavailable or prohibitively expensive, but they do not provide direct access to quantum resources and are therefore limited to problem sizes classically tractable.

**Training & deployment on quantum.**    This setting corresponds to conventional QML. Both training and inference require direct access to quantum hardware, as the model parameters are updated based on measurements from the quantum device (Cerezo et al., 2022; Huang et al., 2022; Biamonte

et al., 2017; Pérez-Salinas et al., 2020; Schuld et al., 2021; Liu et al., 2025b; Khatri et al., 2024; Chen et al., 2020; 2025b). While this approach is the most "native" to quantum computing, it is also the most resource-intensive: the cost of training scales with the number of shots, circuit depth, and optimization iterations, all of which must be executed on a scarce and noisy quantum processor. As a result, this scheme faces significant scalability challenges in the NISQ era.

**Training on quantum, deployment on classical.** A different paradigm is represented by *Quantum-Train* (Liu et al., 2025c; 2024; Chen et al., 2025a; Lin et al., 2024; Chen et al., 2025c; Lin et al., 2025) and related approaches (de Souza et al., 2021; Carrasquilla et al., 2023) such as Quantum Parameter Adaptation (QPA) (Liu et al., 2025d;a). In this scheme, a quantum computer is used during training to generate parameters, embeddings, or compressed representations, which are then deployed in a purely classical model for inference. This design leverages quantum resources where they are most impactful, during training, while avoiding the runtime overhead of quantum hardware in deployment. The trade-off, however, is that the inference stage cannot exploit potential quantum advantages in sampling or generative modeling, since the final model is purely classical.

**Training on classical, deployment on quantum.** Finally, the scheme most relevant to Yomo is to train on classical hardware and deploy on quantum hardware. In this setting, classical simulation is used to optimize the quantum model parameters, which is feasible for medium-scale circuits with efficient simulators such as TorchQuantum or other scalable estimation of expectation value as in (Recio-Armengol et al., 2025a; Kasture et al., 2023; Rudolph et al., 2023). Once trained, the model is executed on a quantum device at inference time, where Yomo's single-shot measurement design becomes highly advantageous. This scheme reduces training cost by avoiding quantum hardware usage during optimization, while still exploiting genuine quantum inference capabilities at deployment. We argue that this hybrid pathway provides a promising balance between practicality and advantage, especially in the NISQ era where inference costs are expected to dominate.

## G  TOWARD PRACTICAL QUANTUM COMPUTING DEPLOYMENT VIA THE YOMO CONCEPT

The Yomo framework demonstrates that QML can be made significantly more practical by rethinking the inference stage, instead of relying on expectation values estimated from a large number of repeated measurements, Yomo can extracts predictions directly from single-shot measurement outcomes. This idea has immediate implications for the cost and accessibility of QML, as it reduces inference overhead by orders of magnitude. More broadly, however, the Yomo concept points toward a general design principle for quantum algorithms, wherever possible, reformulate output mechanisms to minimize dependence on expectation-value estimation.

Many VQAs share the same bottleneck as conventional QML: their objective functions are expressed as expectation values of observables. Examples include the Variational Quantum Eigensolver (VQE) (Kandala et al., 2017), the Quantum Approximate Optimization Algorithm (QAOA) (Farhi et al., 2014), and a wide range of hybrid quantum-classical optimization methods. In all of these cases, the dominant runtime cost arises from repeated circuit executions to estimate expectation values with sufficient statistical accuracy. As system size grows, and as optimization landscapes require repeated evaluations, the number of measurement shots can become prohibitively large, both in terms of wall-clock time and monetary cost on cloud-based quantum hardware.

The success of Yomo in the classification setting suggests a broader research agenda: *can other quantum methods be reformulated to operate in a shot-efficient or even single-shot regime?* For instance, one could envision variants of VQE where single-shot samples are aggregated through tailored loss functions or adaptive rescaling, providing sufficiently accurate gradient signals without the need for thousands of measurements per iteration. Similarly, in QAOA, one could investigate whether problem-dependent mappings allow decision-making or objective evaluation directly from raw bitstring samples, bypassing the need for high-precision expectation estimates.

Exploring these directions requires rethinking the interface between quantum circuits and classical post-processing. Yomo demonstrates that with appropriate probability aggregation and carefully designed loss functions, a model can be trained to produce outputs that are robust even under single-shot measurement. Extending this principle to VQAs would mean designing cost functions,

aggregation strategies, or training procedures that explicitly anticipate the single-shot constraint. In effect, the burden of precision estimation is shifted from the deployment stage to the training or design stage, where it can be managed more efficiently.

We therefore view Yomo not only as a contribution to QML, but as a foundation for a broader paradigm shift in quantum algorithm design. By prioritizing shot efficiency at the output stage, quantum methods can become far more practical to deploy on near-term hardware. This perspective highlights an important research opportunity: to systematically revisit existing variational algorithms, identify their measurement bottlenecks, and seek shot-efficient reformulations inspired by the Yomo concept. Such efforts would directly advance the practical deployment of quantum computing by reducing both the temporal and economic costs associated with measurements, thereby lowering the barrier for real-world applications.

