# OpenReview forum: "You Only Measure Once: On Designing Single-Shot Quantum Machine Learning Models"
_ICLR.cc/2026/Conference — ICLR 2026 Conference Withdrawn Submission_

### Official Review · Reviewer_rkMa · 2025-10-16

**Soundness:** 2
**Presentation:** 3
**Contribution:** 2
**Rating:** 4
**Confidence:** 4

**Summary:**

This paper proposes YOMO, a quantum machine learning (QML) framework that enables single-shot inference. Unlike conventional expectation-value–based QML frameworks, the authors introduce a probability aggregation mechanism that maps bitstrings obtained from single-shot measurements directly to classification labels. In addition, techniques such as probability sharpening are applied in the loss function to further improve performance. Some theoretical analyses are provided to discuss conditions under which YOMO can outperform standard QML approaches. Experiments demonstrate that YOMO achieves better performance than vanilla QML models on MNIST and CIFAR-10 classification tasks when only a few measurement shots are used.

**Strengths:**

- The motivation is clear and well justified. The paper targets a major bottleneck in quantum inference — the high cost of measurement shots.

- The proposed probability aggregation mechanism is an interesting idea, though it introduces several issues that will be discussed below.

- The authors provide theoretical analysis to describe the conditions under which YOMO can achieve lower shot counts than vanilla QML.

- The paper is clearly structured and generally easy to follow.

**Weaknesses:**

- The main concern lies in scalability, which is crucial for realizing quantum advantage. As the authors note, YOMO maps all measurement probabilities, which have a dimensionality of $2^N$ for an $N$-qubit system, to classification labels. This suggests that YOMO effectively requires full state tomography information during training. Therefore, it is highly doubtful that YOMO can be trained within a feasible time for systems with many qubits.

- A second issue concerns the vanilla QML model used for comparison. Mathematically, the training of YOMO seems to rely on a large number of observables to reconstruct the full computational basis, forming a POVM. In that case, the baseline QML model should also employ a comparable set of observables, rather than being restricted to single Pauli product terms.

- The presentation of the theoretical part (Section 5) could be improved. The five theorems may be merged into a more compact form, leaving space for a more detailed discussion of how quantities such as $p$, $\Delta$, and $L$ behave in practice.

**Questions:**

Please refer to Weaknesses 1 and 2.

---

### Official Review · Reviewer_kkqw · 2025-10-26

**Soundness:** 2
**Presentation:** 3
**Contribution:** 2
**Rating:** 2
**Confidence:** 4

**Summary:**

The paper presents YOMO (You Only Measure Once), a quantum ML algorithm that performs classification from a single measurement shot rather than estimating expectation values of observables, as traditionally used in QML. YOMO measures in the computational basis, partitions the possible output bitstrings into K class groups (where K is the number of classes of the classification problem), and defines each class score by aggregating (averaging) the probabilities of the bitstrings in its group. The model is then trained on the full output probability distribution, so at deployment a single measured bitstring maps directly to a class. The authors also provide theoretical bounds showing regimes where YOMO achieves lower error with fewer shots than expectation-value QML, and they report experiments on MNIST and CIFAR-10 where YOMO attains high accuracy with far fewer shots.

**Strengths:**

- The idea of the paper is clear, and, once trained, the model maps a measured bitstring directly to a class, enabling single-shot classification, this is novel relative to standard expectation-based QML measurement.

- The theoretical analysis delineates the regimes in which YOMO outperforms traditional QML methods.

- Experimental results are presented to empirically support the paper’s claims.

**Weaknesses:**

Although the above strengths, I do not think the paper meets the ICLR bar.  The main concern I have with the paper is YOMO’s scalability with qubit count is limited, in particular:


- Training requires access to the entire output probability distribution, which is impractical beyond small systems. This also makes YOMO’s advantage partly unsurprising: expectation values can be computed from the full distribution, whereas standard QML only estimates them and thus needs fewer shots.


- Because it operates on full probability vectors, training becomes extremely slow and expensive as the number of qubits grows.


- The theoretical gains hinge on the single-shot correctness probability p. If, in practice, achieving a high p demands many shots, the supposed shot advantage disappears.


- The authors claim practicality only up to ~35 qubits, there the circuits are still classically simulable, so it’s unclear what advantage a quantum approach offers in the first place

**Questions:**

- Can you train YOMO without reconstructing the entire output distribution, (maybe taking ideas from classical shadows, tensor-network approximations)?

-  If state-vector simulation is required for training, what advantage does a QML approach offer over a purely classical model? Why should a QML method be chosen to solve the task?

---

### Official Review · Reviewer_aGjL · 2025-10-29

**Soundness:** 3
**Presentation:** 3
**Contribution:** 3
**Rating:** 2
**Confidence:** 3

**Summary:**

The paper proposes Yomo (You Only Measure Once), a quantum machine learning framework for classification that reduces the number of shots at inference ideally to just one shot. The main point is to consider the output n-bit bistring as the output after an aggregarion step where classes are defined as subsets of bitsrings.
I am a bit unclear how different this model is from one where the observables are projection operators on exactly those subspaces.
The authors derive shot-complexity bounds identifying when Yomo outperforms expectation-based models. The comparison is validated on MNIST and CIFAR-10, including noisy simulations.

**Strengths:**

This paper constitutes an original and very natural attempt to reduce inference-time shot count. The paper’s setup and objectives are clearly stated, with precise definitions, and the theory is presented through readable theorems. The experimental setup is clearly explained, with informative comparison curves.

**Weaknesses:**

-One thing I was unclear about is if the Yomo approach is scalable. Depending how I read the description it seems that the sharpening method requires a precise estimation of the probabiliites of bistrings? This would make it intractable. On the other hand if the idea is to always be close to quantum states where some probabiliities are large (so calle peaked states), it is entirely unclear how one would train such a model, as peaked circuits are far from dense in the space of most circuits?

-if approximaitons are enough, can this model not be represented by a few copy circuit doing the same thing followed by an observable and in that sense is not actually a different model from standard?

-The authors stress (e.g., in the discussion) that Yomo is not intended to be trained on quantum hardware:
 "It is important to emphasize that Yomo is not intended to be trained directly on quantum hardware.
Instead, its design is particularly well-suited to the setting where training is performed using classical
simulation of quantum states, while deployment takes place on quantum devices. This separation
leverages the flexibility of classical training environments, avoiding the substantial shot cost and
noise challenges of on-hardware optimization." I feel this is a dramatic restriction on the potential of QML which is already under serious question. I am then also confused:  why should we do QML, but not simply use a classical model?
It may be I am misunderstanding things, but this seems like a very serious shortcoming.

-Experiments are on MNIST/CIFAR with small qubit counts. A substantial classical model(two conv–pool layers + a linear layer) precedes the QNN, so the feature extractor may do most of the learning. A fairer comparison would remove or substantially shrink this preprocessing and compare a QNN head vs. Yomo under matched classical capacity. In addition, the QNN baseline uses a fixed set of 10 Pauli observables acting non-trivially only on the first four qubits, which may itself decrease the QNN’s performance

**Questions:**

Some questions were provided in "weaknesses".
As noted, the non-scalability of Yomo’s training seems to negate its potential advantage in shot count. Are there any settings (even artificial) where training can be done in polynomial time in the number of qubits, analogous to the “train on classical, deploy on quantum” paradigm you cite (e.g., arXiv:2503.02934, 2025a), but adapted to Yomo? If yes, could u specify it?

---

### Official Review · Reviewer_ioEH · 2025-10-31

**Soundness:** 2
**Presentation:** 3
**Contribution:** 1
**Rating:** 2
**Confidence:** 5

**Summary:**

This paper introduces a novel Quantum Machine Learning (QML) framework named "You Only Measure Once" (Yomo), designed to significantly reduce the number of measurement shots required for inference. Yomo circumvents the high shot overhead of traditional Parameterized Quantum Circuits (PQCs) by reformulating the model's output mechanism. Instead of relying on expectation values of observables, it makes predictions directly from the probability distribution over the computational basis states. To support this design, the authors introduce a custom loss function featuring "probability sharpening" and entropy regularization, which encourages the model to produce confident and peaked predictions. The paper provides a theoretical analysis that bounds the shot requirements for Yomo and its conventional counterparts, and validates its performance on the MNIST and CIFAR-10 datasets through extensive experiments, including simulations under realistic hardware noise.

**Strengths:**

The paper addresses a critical bottleneck for practical QML: the prohibitive cost of inference due to measurement overhead. The custom loss function, tailored to foster sharp probability distributions, is a thoughtful and complementary contribution.

The authors provide a solid theoretical foundation for Yomo's claimed shot efficiency. They derive upper bounds on the number of shots required for Yomo to achieve a target error probability and formally contrast these with the requirements of conventional expectation-based models.

The experiments are thorough and well-designed. The authors benchmark Yomo against a "Vanilla" QML baseline across various shot budgets on MNIST and CIFAR-10. Crucially, the evaluation extends to the impact of increasing qubit counts and includes an extensive study with simulated noise modeled after four real quantum hardware platforms (Quantinuum, IBM, Google, and IonQ).

**Weaknesses:**

The adaption of this paper is just the replacement of the quantum circuit output: from expectation to concrete measurement results or the distributions, which only allows the inference of a perfect model but no training processes. The training method is then totally unscalable.

The entire training procedure for Yomo relies on having access to the full state vector of the quantum state to compute the complete probability distribution over all 2^n computational basis states. This necessitates the use of a state-vector simulator, a method whose computational cost scales exponentially with the number of qubits,

Although the authors frame their framework as "train-on-classical, deploy-on-quantum," this training paradigm is only feasible for small-scale systems (e.g., fewer than ~30 qubits) that can be tractably simulated on classical computers. This creates a sharp contradiction with the paper's stated goal of "lowering the barrier to practical adoption of QML."  In essence, the proposed method trades an exponential increase in training cost for a reduction in inference cost.

The QML community has proposed various techniques to tackle the high cost of training, including measurable gradients and large-scale tensor network simulations. Although the paper's focus is on inference, the training bottleneck is so pronounced that a discussion comparing Yomo to these works is warranted.

**Questions:**

See weakness

---

### Note · Authors · 2025-11-20

I have read and agree with the venue's withdrawal policy on behalf of myself and my co-authors.